# Development of Design Considerations as a Sustainability Approach for Military Protective Structures: A Case Study of Artillery Fighting Position in South Korea

**Kukjoo Kim** [1,2] **and Youngjun Park** [1,*]

1   Department of Civil Engineering and Environmental Sciences, Korea Military Academy, Seoul 01805, Korea; klauskim@ufl.edu
2   Nuclear·WMD Protection Research Center, Korea Military Academy, Seoul 01805, Korea
*   Correspondence: parky@mnd.go.kr

**Abstract:** Republic of Korea (ROK) military installations are scattered across South Korea, but there is a higher concentration of fortifications in the demilitarized zone (DMZ) and eastern and western coastlines. These facilities range from relatively small structures, such as individual and artillery fighting positions, to large buildings, such as ammunition depots and command posts. These military installations have a significant thickness of concrete members to provide a high degree of protection against bombs and projectiles. The Korean military will carry out the integration and dismantling of these protection facilities over the next ten years through the Army transformation plan. Such large-scale construction projects have an impact on the environment in terms of the carbon footprint, because building construction and operations account for 36% of the world's energy use and 40% of energy-related carbon dioxide ($CO_2$) emissions. It is very important to reduce the concrete materials and reinforcement steel during protective structure construction near the DMZ, which is now recognized as one of the most well-preserved areas in the world. In this study, new sustainable design considerations that allow elasto-plastic or plastic design of concrete elements were evaluated using a case study of an artillery fighting position. The new sustainable design considerations were developed on the basis of mission, enemy, terrain and weather, troops and support available, time and civil considerations (METT + TC) within the context of the current battle situation, as well as protection against near misses. From this study, it was found that new sustainable design considerations provide a reasonable degree of protection that permits good construction practices and maximum structural stability with minimum amount of materials. It was also found that if the new design procedure is used to replace 1000 artillery positions through the Army transformation plan, the $CO_2$ emissions can be reduced by 476,582.4 tons and the cost reduced by USD 23,829,120.

**Keywords:** degree of protection; impact damage; blast wave; sustainable design consideration; elasto-plastic design; $CO_2$ emission

## 1. Introduction

### 1.1. Background

The design of protective structures is an important factor not only for military construction but also for civilian sectors. As the threat of enemy's weapons of mass destruction increase, protective structure design becomes a common problem for military, civil, and industrial facilities. Currently, there is little information (including experimental data regarding bombs, projectiles, and atomic bomb

blasts, etc.) and design procedures available to serve as a design guideline for such protection structures. In conventional works, the maximum degree of protection has been used on the basis of a 00-pound GP bomb detonating at a distance of 00 m (restriction on disclosure due to military secrets). These design criteria produce a structure which is able to sustain a given loading condition within the limits of elastic strain, which requires a significant amount of concrete. Reducing the amount of concrete used in a construction project is very important in terms of sustainability awareness and green planning [1]. The International Energy Agency and United Nations (UN) Environment Programme stated that building construction and operations accounted for 36% of the world's energy use and 40% of energy-related carbon dioxide ($CO_2$) emissions in 2017 [2].

More specifically, Pacheco-Torgal et al. described that concrete and reinforcement steel account for about 65% of building greenhouse gas (GHG) emissions, 40% of which is $CO_2$ emissions from concrete [3]. It is noted that the mean embodied carbon dioxide ($ECO_2$) for building is 340 kg-$CO_2$/m$^2$, of which the structure accounts for about 60% [4]. This means that reducing the $ECO_2$ in the structure frame directly reduces the GHG emissions. Additionally, in terms of the carbon footprint, it is very important to reduce the concrete materials and reinforcement steel during construction projects [5–9].

Recently, the Korean military has formulated plans to carry out the integration and dismantling of these protection facilities through the Army transformation plan over the next ten years. As military protective structures are concentrated at the border, these enormous concrete structures adversely affect the environment, particularly in the demilitarized zone (DMZ), which is now recognized as one of the most well-preserved areas in the world.

To identify the appropriate degree of protection, a design process must consider the weapon effects and dynamic factors pertaining to mission, enemy, terrain and weather, troops and support available, time and civil considerations (METT + TC) [10] within the context of the current battle situation. It must be considered that structure members can resist dynamic loads under relatively large plastic deformation. Such local overstresses in the member, or even some failures, should not seriously impair the overall structure. Some protective structures, such as artillery fighting positions, require protective ability only once. If the protective structure design process ignores the METT + TC factors, it produces structural members with massive thickness. As a large amount of concrete materials is consumed, the construction of these structures has a direct impact on the natural environment. Therefore, it is important to reduce the use of concrete and non-renewable materials during construction works.

In this study, new protective structure design considerations were developed to improve the resistance of structure members, resulting in large plastic deformation. The new design considerations evaluate the amount of concrete that was saved while providing an appropriate degree of protection by using finite element (FE) analysis as a case study.

*1.2. Objectives and Scope*

The primary objective of this study was to develop new sustainable design considerations for protective structures, using the Delphi technique on the basis of METT + TC factors within the context of the current battle situation. Then, after applying the proposed design consideration to the case project, the $CO_2$ emission and cost reduction corresponding to the concrete savings were analyzed. To do this, a three-dimensional FE analysis was conducted to assess the potential performance of the artillery fighting position as a case study in South Korea.

## 2. Protection against Conventional Weapons

For the purpose of protection against weapons, the protective structures may be classified into two general groups: those which provide protection against (1) the impact of a weapon's penetration and (2) the blast of a weapon's explosion. Penetration is caused by weapons such as projectiles fired from guns, conventional bombs with a charge-to-weight ratio smaller than 20%, rockets, and guided missiles. Explosion blast is caused by weapons such as high explosive or conventional bombs with a charge-to-weight ratio higher than 20%. For the purpose of structural analysis, a weapon's

impact causes severe local damage, while the weapon's blast causes overall damage of relatively less severity [11,12].

When a bomb or projectile strikes a concrete member, there is the formation of an irregularly shaped crater and considerable cracking in the opposite side of the slab. The severity of such cracking decreases as the concrete thickness increases. Because of the inherent low tensile strength of concrete, both faces of the slab tend to rupture with the reflected shock wave in the impact face and the propagated wave in the opposite face. Design information about the weapon system and condition of protection to be provided is necessary. In most cases, the desired level of protection of a structure differs [13–15]. For example, if the building is to be located near the border between nations, within the range of army artillery, the required protection of the exposed walls would be the loading due to an armor piercing (A.P.) type projectile. In contrast, the design of the roof of the building would consider the loading of and A.P. type bomb released from a carrier plane.

In many cases, the functional importance of a protective building, its size and the thickness of structural members is larger so as to provide a certain degree of lateral and overhead protection against blast and fragments of a bomb. In the South Korea Army, a reasonable degree of protection has been developed on the basis of a 00-pound GP bomb detonating at a distance of 00 m. The thickness of structural members resulting from this consideration only permits the induced stresses of structure elements to remain in the elastic range. However, the blast loading on a protective building caused by a high explosive detonation in a bomb depends on the peak pressure and the impulse of the incident and dynamic pressures. For the analysis of structures under dynamic loading, such as blast loading, the analysis of inertial force and kinetic energy is required, as the applied load changes rapidly with time, as shown in Figure 1 [16].

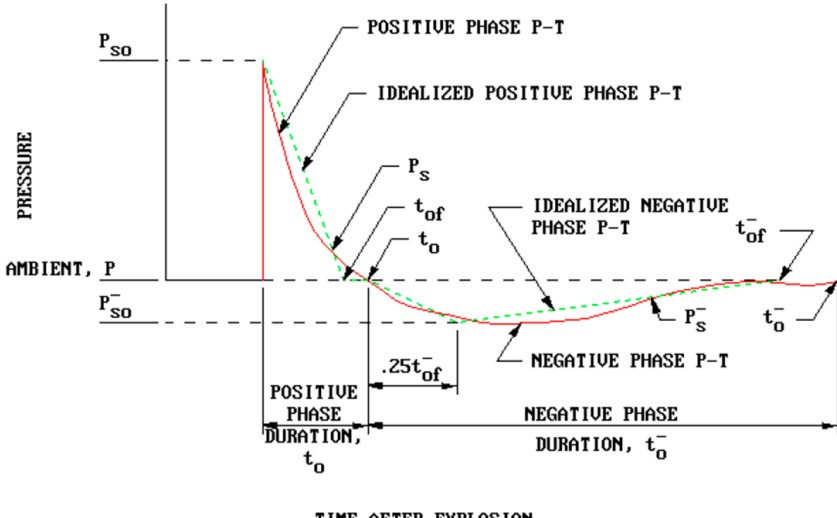

**Figure 1.** Idealized pressure-time curve of a blast wave.

For design purposes, the effects of the inertial force in the equation of dynamic equilibrium and kinetic energy in the equation of energy conservation related to the mass of the structure must be considered. The response of a concrete element in a protective structure can be defined as ductile or brittle structural behavior. In the ductile mode of response, large inelastic deflections without complete collapse occur in the structure element, while partial failure or total collapse of the element occurs in the brittle mode [17]. If the ductile behavior is selected for an element of protective structures in the current design consideration, there can be savings in concrete materials within the desired level of protection. The flexural action of a reinforced concrete member was demonstrated by the resistance–deflection curve shown in Figure 2 [16].

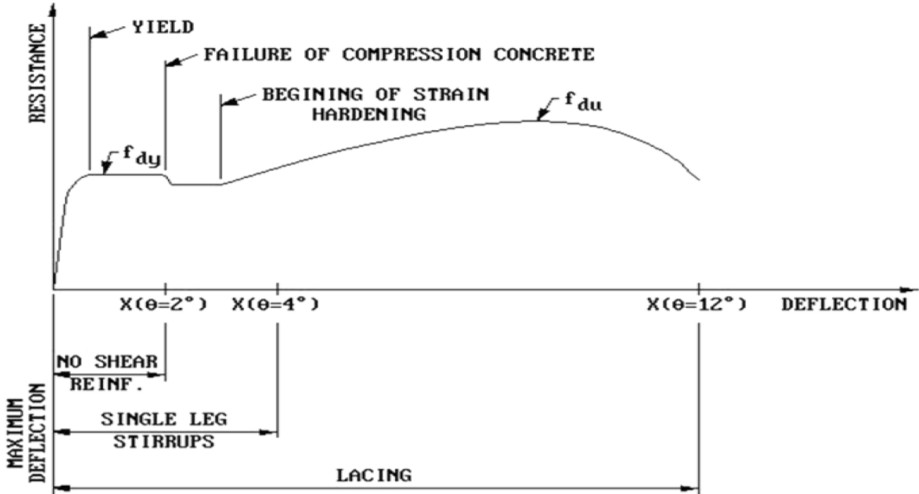

**Figure 2.** Resistance-deflection curve for flexural response of concrete elements.

The magnitude of stresses produced in the protective structure responding in the plastic range cannot be directly related to the strain. The average stress over portions of the plastic range can be determined by relating this average stress to the deflection of the element defined in terms of the angular rotation at the supports. Therefore, the elasto-plastic or plastic design considerations of concrete protective structures must be considered [18] in the current Army protective structure design standard to use maximum protection capability of concrete elements, and sustainable and economical design approaches.

## 3. Development of New Design Considerations as a Sustainable Approach

The protection standard of the Republic of Korea (ROK) armed forces comprises four stages, each of which is determined based on comprehensive considerations for the threat of the enemy and the protective capability from the aspect of military operations and for the purposes from the aspect of facilities. Once a degree of protection is set, the corresponding protection level of a structure is determined based on blast loads. In general, protection levels according to protection degrees represent the thresholds of the displacement ductility factor and rotation angle that are proposed by the Unified Facility Consideration (UFC) 3-340-02 [16]. Table 1 presents the permissible limits of rotation angle for brittle materials such as concrete at each protection level.

**Table 1.** Design consideration of the protective facilities in UFC 3-340-02.

| Protection Level | Construction Method | Damage Aspect | Maximum Support Rotation Angle |
|---|---|---|---|
| A | Elastic design (Working stress design) | Microcrack | 0–2° |
| B | Carbon design | Protection for human lives (Crack, crush) | 2–6° |
| C | Plastic design (limit design) | Severe collapse (Separation of concrete from the reinforcement bar) | 6–12° |

The protection levels of Table 1 are different concepts from the degrees of protection. The protection levels are distinguished based on design concepts. In the case of the ROK armed forces, when the protection degree of a protective facility is determined, an elastic design corresponding to the protection level A is adopted.

For the design of a protective structure, it is necessary not only to analyze severe dynamic loads comprising blended impacts of blast waves and fragments, but also to examine various and complex battlefield conditions where projectiles might directly blast and penetrate the structure. However, the protective degrees currently in use in the ROK armed forces are still grounded on the dated concept of protection focusing on the thickness of heavy-weight structures.

This study aims to propose guidelines for determining bullet/explosion-proof degree of protection, whose application ranges from high-tech precision guided weapons to the artillery strength for pinpoint strike, and to examine guidelines for future revisions in the area of protection in the standard of defense and military facilities. To achieve these goals, the Delphi technique was used to accurately reflect objective opinions of experts from government, military, and private sectors. Based on the opinions collected, the guidelines for determining degrees of protection were derived by horizontally and vertically synthesizing key words that were extracted from the Korea Army innovation assessments and the innovation school of the Korea Army Research Center for Future and Innovation (KARCFI). To achieve a fair and even distribution, a group of 21 experts (7 civilian experts, 7 government officials, and 7 servicemen) was organized. All the experts were experienced in defense and military facilities.

After organizing the expert group, we conducted several rounds of survey (first round with open-ended questionnaires and second to fourth rounds with closed-ended questionnaires). Based on the survey, we derived considerations for setting protection degrees. In particular, the Shapiro–Wilk normality test was performed to quantify the agreement between each panel during the second to fourth rounds. Then, a factor analysis was performed to identify common features of considerations in each factor [19–21]. The result was summarized into the five tactical considerations (METT + TC). Then, the innovation school and assessments for future battlefield environment led by KARCFI extracted essential considerations for setting protection degrees as key words and combined them horizontally and vertically. Consequently, the considerations identified for the protection standard of military facilities include the following six factors: wartime/peacetime mission; omnidirectional threat; stability and resilience of troops; geology and weather; threat detection, alert, reaction and recovery time; military-private combined factor. The design process checklist is shown in Table 2, avoiding excessive design and ensuring the desired performance while considering future diversified battlefield environments and weapon systems. The highest requirements for each item in Table 2 are selected as the final degree of protection and protection level. Table 2 shows an example of determining the degree of protection and protection level for artillery positions.

**Table 2.** Example of the new design process checklist for artillery fighting position.

| Classification | Considerations for the Protection Standard of Military Facilities | | Degree of Protection | | | Protection Level | | |
|---|---|---|---|---|---|---|---|---|
| | **6 Factors** | **Detailed Items** | **Direct Hit** | **Contact Explosion** | **Near Misses** | **A** | **B** | **C** |
| M (Mission) | Wartime/peacetime mission (4 items) | Peacetime mission | | | ● | | | ● |
| | | Wartime mission | | | ● | | | ● |
| | | Importance of mission | | | ● | | | ● |
| | | Camp protection plan | | | ● | | | ● |
| E (Threat of the enemy) | Omnidirectional threat (4 items) | Present threat (Tactic, arrangement, organization, and activity of enemy) | | | ● | | | ● |
| | | Potential threat | | | ● | | | ● |
| | | Transnational threat | | | ● | | | ● |
| | | Non-military threat | | | - | | | - |

**Table 2.** *Cont.*

| Classification | Considerations for the Protection Standard of Military Facilities | | Degree of Protection | | | Protection Level | | |
|---|---|---|---|---|---|---|---|---|
| | 6 Factors | Detailed Items | Direct Hit | Contact Explosion | Near Misses | A | B | C |
| T (Available troops) | Stability and resilience of troops (9 items) | Type of troops | | | ● | | | ● |
| | | Location and capacity of protective facility | | | - | | | - |
| | | Possession of explosive weapons | | | ● | | | ● |
| | | Size of troops | | | ● | | | ● |
| | | Smart safety system | | | - | | | - |
| | | Possession of reserve facility | | | ● | | | ● |
| | | Loss of lives caused by the destruction of a facility | | | ● | | | ● |
| | | Damage recovery capacity (each unit and higher command) | | | ● | | | ● |
| | | Threat response capacity | | | ● | | | ● |
| T (Terrain and weather) | Geology and weather (5 items) | Natural terrain | | | ● | | | ● |
| | | Man-maid terrain | | | ● | | | ● |
| | | Avenue of maneuver inside and outside camp | | | - | | | - |
| | | Vegetation | | | ● | | | ● |
| | | Weather change | | | - | | | - |
| T (Available time) | Threat detection, alert, reaction, and recovery time (6 items) | Threat detection time | | | - | | | - |
| | | Threat alert time | | | - | | | - |
| | | Evacuation time | | | - | | | - |
| | | Operation time of protective facility | | | - | | | - |
| | | Time for neutering threat | | | ● | | | ● |
| | | Damage recovery time | | | - | | | - |
| C (Civil factor) | Military-private combined factor (6 items) | Comprehensive land development plan | | | ● | | | ● |
| | | Military-civil relationship | | | ● | | | ● |
| | | Civil complaint | | | ● | | | ● |
| | | Importance of facility | | | ● | | | ● |
| | | Core technology for establishing protective facilities | | | - | | | - |
| | | Civilian protection capacity | | | - | | | - |
| Overall judgment | | | Close strike of enemy artillery | | | Protection level C | | |

● indicates applicable, - indicates not applicable.

## 4. A Case Study for Artillery Fighting Position Using Finite Element Analysis

### 4.1. Setting the Protection Degree and Level

When designing a protective structure, it is necessary to consider the dynamic loads of blast waves and impacts of fragments caused by the explosion of a high-energy bomb or a missile. Regarding such dynamic load, the characteristics of a weapon as a means of strike need to be closely examined. A case study for artillery positions in the front-line area was performed by applying the design factors of Table 2. Specifically, the standard type of the existing artillery positions and a new artillery position designed with the new design factors were comparatively evaluated through a FE analysis.

The major threat issue of the new artillery position is not the direct strike of enemy artillery, but rather the protection against blast waves and fragments caused by close explosions. METT-TC of the artillery troops being considered, the protection level was set to Level C, as the fire-and-displace is expected under the enemy's counter-artillery fire. Through the analysis, the protection degrees and levels that were ultimately desired were derived, as shown in Table 2.

### 4.2. Evaluation of Protective Performance through Numerical Analysis

This study performed a FE analysis to identify the dynamic behavior characteristics of an artillery position under blast waves. It was assumed that 115 kg of TNT, the maximum explosion in enemy's weapon, was exploded 7.6 m away from the artillery fighting position. This is the result of taking into account the accuracy of enemy artillery weapons without using guided weapons during artillery battles.

A numerical analysis mode was developed by using ANSYS AUTODYN®. This program was developed by the Institute for Defense Analysis. It is a very useful tool for solving the ductility issue between fluids and solids through the coupling of Lagrange and Euler Solvers in solid mechanics. For a non-linear dynamic analysis of a structure, a reinforced concrete element was constructed using an explicit FE method. A standard artillery fighting position with a wall length and height of 4500 and, 700 mm, respectively, was selected as the target structure of the analysis. As for the wall thickness, five cases of 300, 350, 400, 450, and 500 mm were considered.

As for the material properties of the concrete wall, as presented in Table 3, the ordinary concrete and reinforcement bar presented tensile strengths of 24 and 400 MPa, respectively. The minimum reinforcement ratio was 0.00306.

**Table 3.** Material properties used infinite element (FE) models.

| Classification | Wall Thickness (mm) | Reinforcement Bar Diameter | Reinforcement Bar Spacing | Number of Reinforcement Bars | Reinforcement Ratio |
|---|---|---|---|---|---|
| CASE 1 | 300 | HD16 | 300 | 15 | 0.003161 |
| CASE 2 | 350 | HD16 | 250 | 18 | 0.003070 |
| CASE 3 | 400 | HD19 | 300 | 15 | 0.003119 |
| CASE 4 | 450 | HD19 | 250 | 18 | 0.003213 |
| CASE 5 | 500 | HD22 | 300 | 15 | 0.003216 |

As illustrated in Figure 3, the detonation point was 7.6 m away from the artillery fighting position. The explosive was TNT with a density of 36.675 kg/m per unit length in the z-direction.

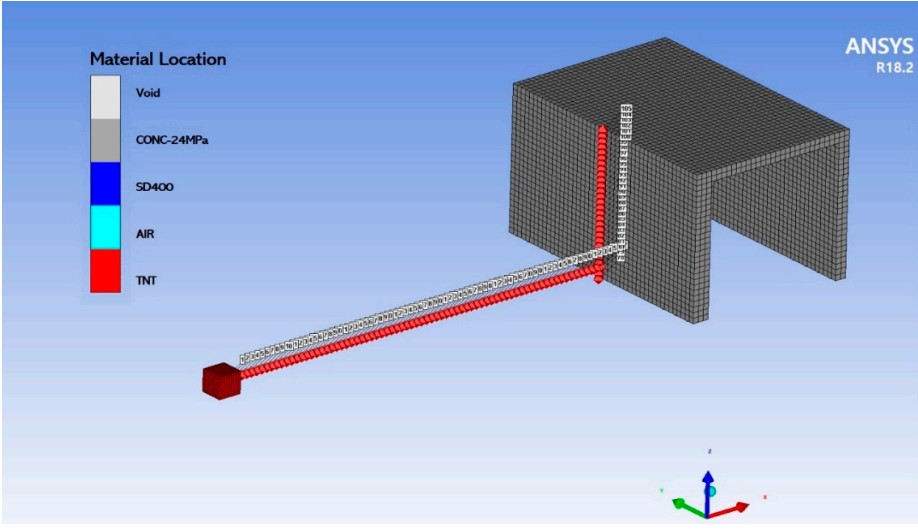

**Figure 3.** FE model developed.

### 4.3. Numerical Analysis Result of Protective Performance

Figure 4 shows the impacts of blast waves on the structure over time. As the March front was lower than the height of the structure, the structure was affected by non-uniform pressures

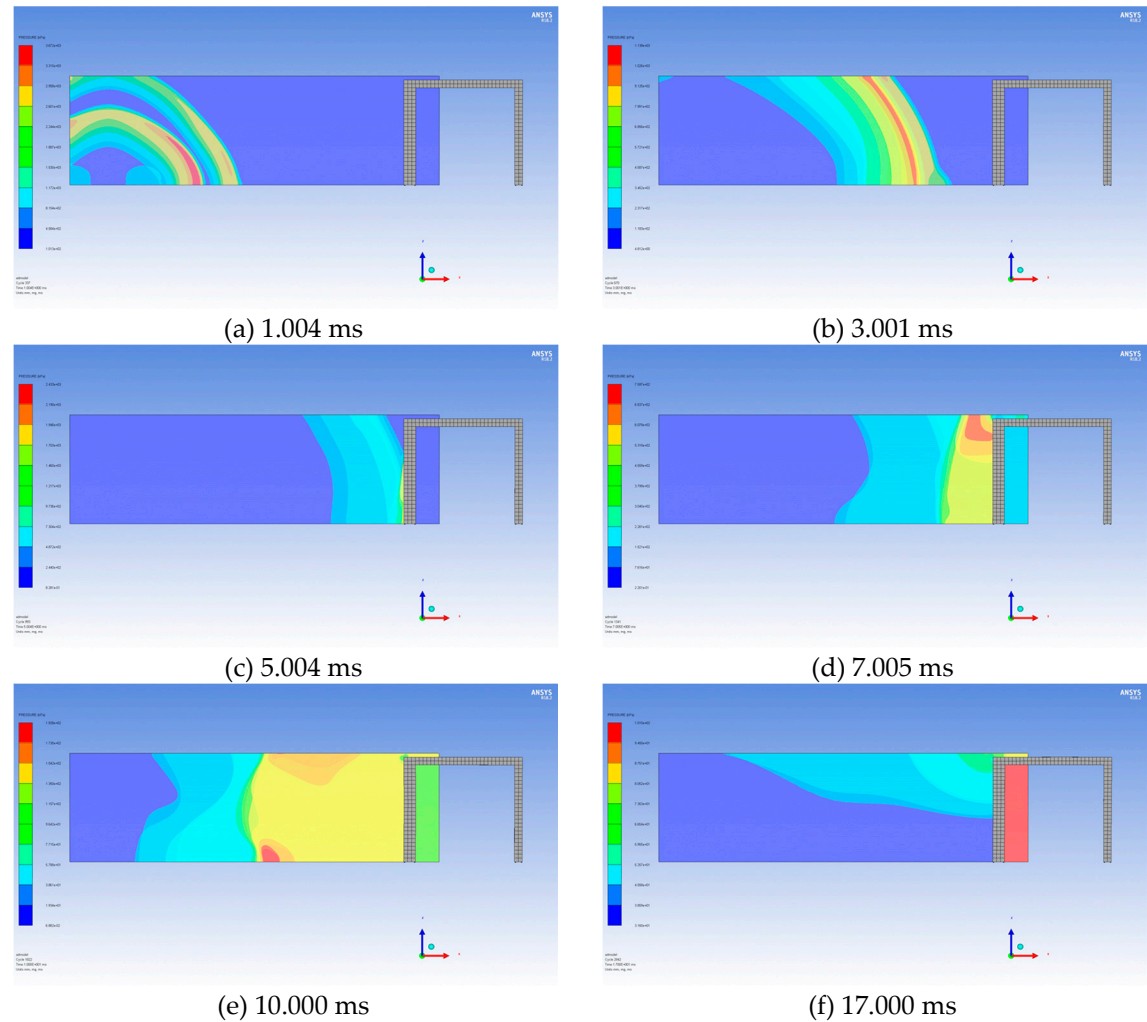

(a) 1.004 ms

(b) 3.001 ms

(c) 5.004 ms

(d) 7.005 ms

(e) 10.000 ms

(f) 17.000 ms

**Figure 4.** Impact of blast waves on the structure over time.

Figure 5 shows the pressures of blast waves and the displacement of the structure wall over time.

Protection degrees and levels desired for the artillery fighting position can be expressed by the maximum displacement and rotation angle of the wall. Table 4 presents protection levels for each case of wall thickness.

As shown in Table 4, the dynamic analysis of the reinforced wall revealed that a sufficient level of protective capacity could be secured, even if the wall thickness was reduced to 300 mm. In other words, if the current design of protective facility reflecting only the elastic displacement of reinforcement structure is replaced by the elasto-plastic design considering the protection levels for each METT + TC factor, as presented in Table 2, protective structures would be more economical and sustainable.

**Table 4.** Maximum displacement and rotation angle according to wall thickness.

| Classification | CASE 1 | CASE 2 | CASE 3 | CASE 4 | CASE 5 |
|---|---|---|---|---|---|
| Maximum Displacement (mm) | 18.5800 | 12.0834 | 12.08 | 9.8921 | 8.3938 |
| Rotation angle (°) | 1.9600 | 0.2564 | 0.2563 | 0.2099 | 0.1781 |

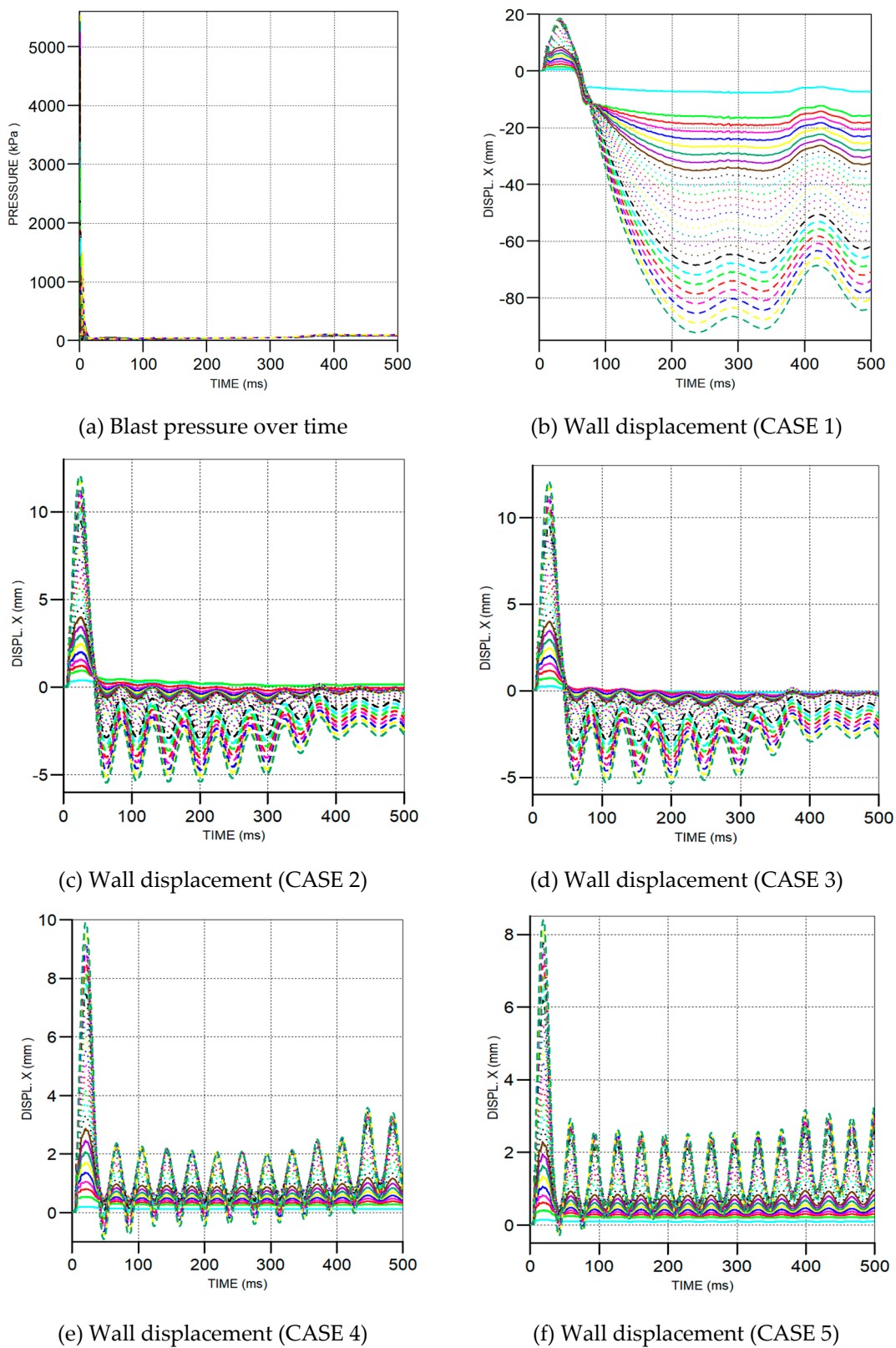

(a) Blast pressure over time

(b) Wall displacement (CASE 1)

(c) Wall displacement (CASE 2)

(d) Wall displacement (CASE 3)

(e) Wall displacement (CASE 4)

(f) Wall displacement (CASE 5)

**Figure 5.** Blast pressure over time and displacement according to wall height.

## 4.4. $CO_2$ Emission Reduction Effects

When using the new design consideration proposed in this paper, the effect of concrete savings should be confirmed. Table 5 shows the calculation results of the $CO_2$ emissions for concrete saved

by the new design procedure as a sustainable approach, when 1000 artillery positions are replaced through the Army transformation plan. When the unit $CO_2$ emissions of ready-mixed concrete used are 3.152 ton-$CO_2$/ton [22], the $CO_2$ emissions from artillery position project Army planed can be reduced by approximately 476,582.4 tons, which is equivalent to 40% of the project. When Korean carbon transaction price of USD 50/ton-$CO_2$ [23] is applied, the total cost savings of USD 23,829,120 can be calculated. Therefore, if the new design consideration proposed in this study are applied to the entire military protective structure projects, greater cost saving and reduction in $CO_2$ emissions are expected.

**Table 5.** Calculation of $CO_2$ emission reduction effect.

| Description | Quantity (ton) | Unit $CO_2$ Emission (ton-$CO_2$/ton) | Amount (ton-$CO_2$) |
|---|---|---|---|
| Existing design procedure (A) | 378,000 | 3.152 | 1,191,456.0 |
| New design procedure (B) | 226,800 | 3.152 | 714,873.6 |
| Reduction effect (A–B) | 151,200 | | 476,582.4 |

## 5. Conclusions

Reducing the amount of concrete used in construction project is very important in terms of sustainability awareness and green planning to reduce carbon and climate change risk globally. The concrete material and reinforcement steel account for about 65% of building greenhouse gas (GHG) emissions, 40% of which is $CO_2$ emissions from concrete. Therefore, green building planning is very important during construction projects. However, the Korean military's design concept does not take full advantage of the features of reinforced concrete structures, resulting in excessive design. The protection scheme of ROK armed forces consists of four stages. In this scheme, protection degrees are set based on relative protection capabilities against particular weapon systems. Furthermore, the protection degrees established require the protection level A, corresponding to the concept of elastic design. Accordingly, no effective protection using the behavior characteristics of structures for weapon systems is provided. As a result, the degrees of protection currently in use in the ROK armed forces are still grounded in the dated concept of protection focusing on the thickness of heavy-weight structures. This study derived the protective design considerations necessary for future protective facilities to avoid excessive design and to secure a desired level of protection performance. In addition, this study also conducted a Delphi process by organizing a group of experts from the government, military, and private sectors. The result of the Delphi method was combined with the design considerations for protective facilities that were derived by the innovation school of KARCFI and innovation consulting. Thus, sustainable design considerations for protective facilities were obtained.

Using the above considerations, an FE method was performed for the protection performance of the standard artillery position widespread in the frontline area. The protection against close explosion was determined based on METT + TC of the artillery position in each protection degree. A dynamic analysis of a reinforced structure showed that the elasto-plastic design could produce a more sustainable structure.

So far, protective structures have been regarded as heavy-weight structures with thick walls. However, if the new design considerations developed in this study are applied, more economical and sustainable protective facilities can be constructed. In particular, the case study revealed that thousands of artillery positions in the frontline area and DMZ can reduce wall thickness. For instance, if the new design procedure is used to replace 1000 artillery positions through the Army transformation plan, the $CO_2$ emissions can be reduced by approximately 476,582.4 tons, which is equivalent to as cost of USD 23,829,120. Therefore, if the new design consideration proposed in this study is applied to the entire military protective structure projects, greater cost saving and $CO_2$ emissions are expected. It confirms that it is possible to provide sustainable protective facilities while satisfying the operational requirements for such artillery positions.

**Author Contributions:** Conceptualization, Y.P. and K.K.; Methodology, K.K.; Software, Y.P.; Validation, Y.P. and K.K.; Formal Analysis, K.K.; Investigation, K.K.; Resources, Y.P.; Data Curation, K.K.; Writing-Original Draft Preparation, K.K.; Writing-Review & Editing, Y.P.; Visualization, K.K.; Supervision, Y.P.; Project Administration, Y.P.; Funding Acquisition, Y.P. All authors have read and agreed to the published version of the manuscript.

**Funding:** This research was supported by a grant (18SCIP-B146646-01) from the Korea Agency for Infrastructure Technology Advancement.

**Acknowledgments:** This work was supported by research fund of the Korea Agency for Infrastructure Technology Advancement. The ROKA Nuclear WMD Protection Research Center at Korea Military Academy is gratefully acknowledged for providing the support that made this study possible.

**Conflicts of Interest:** The authors declare no conflict of interest. The funders had no role in the design of the study; in the collection, analyses, or interpretation of data; in the writing of the manuscript, or in the decision to publish the results.

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
