# Peer review of "Development of Design Considerations as a Sustainability Approach for Military Protective Structures: A Case Study of Artillery Fighting Position in South Korea"

_sustainability, doi:10.3390/su12166479_

Round 1

Reviewer 1 Report

Please use the MDPI style across the whole manuscript.

Line 29. I suggest to include some other key words here, such as 'elasto-plastic' or/and 'METT-TC'.

Line 131. I suggest to expand the 'ROK' in this passage again (regardless of the Abstract).

Line 207. Check the form of record: AUTODYN or maybe ANSYS AUTODYN® 16/ANSYS AUTODYN 16.

Line 266. Please provide the scale bar into the drawing. The color legend stacked bar plot is clear, but the description (font size) to seems to be to small.

Author Response

The authors would like to acknowledge and thank all reviewers for providing invaluable and constructive review comments in relation to this article. These comments are sincerely appreciated and have clearly made a positive impact on the quality of the paper. Based on the reviewers’ suggestions, changes were done to the paper to make it hopefully clearer and more understandable.

Please refer to the attached document and revised manuscript.

Reviewer 2 Report

Thank you for the opportunity to review your paper. I found the subject matter and applied method very interesting. I think there is value in determining the sustainable design of protective structures. 

I believe your paper needs major improvements and addition to be considered for publication in this journal. As it is written, this is not really a sustainability paper; rather, it is more suited for a blast effects or performance of constructed facilities journal. 

I have three major areas for improvement: 

  1. You need to discuss the motivation behind the paper. You mention that there are thousands of the structures in the ROK but do they need to be replaced? If so, how many and how soon? A more sustainable design is only important if you are required to build a significant number of these structures.
  2. I struggle to believe that a lower level of protection is desired in an age of more powerful and higher accuracy weapons. Without seeing your Delphi questionnaire, it is difficult to determine if a lower level of protection is actually desired. Additionally, the results of a Delphi study do not necessarily equate to new policy. Has ROK force protection guidance changed? 
  3. How does this paper focus on sustainability? There is no real discussion or analysis regarding how the design change will impact sustainability. The only real mention of sustainability is that thinner walls will result in less concrete. Given the scope of this journal, I believe the analysis should focus on the resulting sustainability that could be achieved. 

In addition to the major issues above, I have several other content and formatting issues:

  1. Line 16 - unclear what is meant by "have impact on the DMZ environment"
  2. Line 18 (and others) Why is the bomb size and detonation distance not specified? If it is classified or FOUO, then I would state that fact, not leave it as "00"
  3. Line 23 - either here or in the introduction, you should cite and explain METT+TC (in terms of the fact that it is a common military planning tool).
  4. Do not use ampersands in the paper
  5. Line 25 - "massive" should be "mass"
  6. Line 54 - "structure" should be "structural"
  7. Line 80/81 - "irregular shape" should be "irregularly shaped"
  8. Line 101 (and others) - few instances where there is not a space  between text and citation number
  9. Format tables as they would normally appear in academic journals, not with borders around all cells
  10. Line 166/167 - "1st", "2nd", "4th" should be spelled out
  11. Line 176 - spell out "six" instead of "6"
  12. Line 176 - your list of six factors is confusing. I recommend using semicolons to separate them out (as there are commas in between items in the list)
  13. Table 2 - I see no value in this table because all items have the same degree of protection and protection level. I also find it very difficult to believe that government and military members would advocate for lower levels of protection. 
  14. Line 206 - explain the logic behind your choice of blast size and distance from the facility. If this is not a standard design explosive, you probably should analyze additional blast charge weight and standoff distance combinations. 
  15. Could you not also consider different facility shapes/sizes or methods of construction? i.e. buried, semi-buried? This could certainly lead to more sustainable results. 
  16. Line 223 does not agree with line 206 regarding 7.6 m away versus 7.6 meters above the ground. 
  17. Line 281 - similar to one of my major points above, do the buildings need to be replaced? If not, how is this a more sustainable solution? 

Author Response

(The authors gave the same response as above.)

Round 2

Reviewer 2 Report

Thank you for the opportunity to re-review your paper, and for the changes that you have made. I believe the added sections brought the paper within the scope of the journal, although I still feel that it may have been better suited in a different journal. That opinion aside, I have just a few minor comments and suggestions. 

Line 16/58 - should this be "Korean" military instead of "Korea" military? 

Line 17 - suggest "over the next ten" instead of "for the next ten"

Line 31/32 and where appropriate in the body of the paper - recommend using significant figures (for 476,582.4 tons and the $23,829,120)

Line 32 - "Through a case study" is not a sentence. I believe this was intended to be the completion of the previous sentence.

Line 211 - "Since it does not..." is a fragment. Please revise.

Line 213 - I realize that you added this because of my first-round comment. It does not need to be included in the paper. Rather, I just have a hard time basing your conclusions on a single case study with a single design. How can you confirm that your design would be sufficient if you only used one explosive charge weight and one standoff distance?

Line 217 - should "non-liner" be "non-linear"?
